# Essential Oil Content, Composition and Free Radical Scavenging Activity from Different Plant Parts of Wild Sea Fennel (*Crithmum maritimum* L.) in Montenegro

**DOI:** 10.3390/plants13142003

**Published:** 2024-07-22

**Authors:** Ljubomir Šunić, Zoran S. Ilić, Ljiljana Stanojević, Lidija Milenković, Dragana Lalević, Jelena Stanojević, Aleksandra Milenković, Dragan Cvetković

**Affiliations:** 1Faculty of Agriculture, University of Priština in Kosovska Mitrovica, 38219 Lešak, Serbia; ljubomir.sunic@pr.ac.rs (L.Š.); lidija.milenkovic@pr.ac.rs (L.M.); dragana.lalevic@pr.ac.rs (D.L.); 2Faculty of Technology, University of Niš, Bulevar Oslobodenja 124, 16000 Leskovac, Serbia; ljiljas76@yahoo.com (L.S.); jelena_stanojevic@yahoo.com (J.S.); aleksandra.milenkovic@student.ni.ac.rs (A.M.); dragancvetkovic1977@yahoo.com (D.C.)

**Keywords:** sea fennel, plant parts, essential oil, monoterpene hydrocarbons, limonene, antioxidants

## Abstract

This study was conducted to determine the sea fennel essential oil (SFEO) yield, composition, and antioxidant activity of leaves, stem, inflorescences, and umbels from seeds of wild sea fennel (SF) (*Crithmum maritimum* L.) from the Montenegro coast. The chemical composition of isolated essential oil was determined by GC/MS and GC/FID analyses. The antioxidant activity was determined using the DPPH assay. The maximum SFEO yield was found in umbels with seeds (4.77 mL/100 g p.m.). The leaves contained less EO (0.52 mL/100 g p.m.) than immature inflorescence (0.83 mL/100 g p.m.) The minimum EO content was found in the stem (0.08%). Twenty components were isolated from SFEO leaves, twenty-four from inflorescence, thirty-four components from the stem, and twenty-one components from umbels with seeds. Limonene (62.4–72.0%), γ-terpinene (9.5–14.0%), α-pinene (1.4–5.8%), and sabinene (1–6.5%) were found to be the main components of the SFEO from monoterpene hydrocarbons as dominant grouped components (86% to 98.1%). SF plant parts showed differences in chemical profiles, especially in specific and low-represented ingredients. (*E*)-anethole (4.4%), fenchone (0.5%), and *trans*-carveol (0.2%) were present only in umbel with seeds, while the β-longipipene (0.5%), (*E*)-caryophyllene (0.5%), and (2*E*)-decenal (0.2%) were found only in the stems. The degree of DPPH radical neutralization increased with incubation time. The SFEO isolated from the stems showed stronger antioxidant activity during the incubation times of 20 and 40 min (EC_50_ value of 5.30 mg/mL and 5.04 mg/mL, respectively) in comparison to the SFEO isolated from the other plant parts. The lowest antioxidant activity was obtained with the SFEO leaves (155.25 mg/mL and 58.30 mg/mL, respectively). This study indicates that SFEO possesses significant antioxidant activities and is animportant component in the food and pharmaceutical industries. It is important to preserve the existing gene pool and biodiversity with rational use SF for the extraction of high-quality essential oils.

## 1. Introduction

Sea fennel (SF) or ‘samphire’ (*Crithmum maritimum* L.) is a wild perennial halophyte spontaneously growing across the Atlantic seaside and in many countries on the Mediterranean coast, adapted to survive in difficult living conditions of salty and rocky surfaces [1]. Sea fennel is used as an edible plant for culinary purposes and presents a good source of nutraceutical compounds [2]. It is spread over the entire coastal belt of Europe, even in colder regions. Sea fennel grows on Montenegro’s cost in the immediate vicinity of the sea, where it grows from rocks and sand. It is a perennial halophytic plant of the Apiaceae family. It is the only plant in the *Crithmum* genus. The succulent leaves are thick, deeply cut, fleshy, and gray–olive in color. Inflorescences are umbels with white greenish flowers; fruits are schizocarps with two partitions with numerous seeds [3]. They bloom from July to September. *C. maritimum*, with a halophytic nature, represents an alternative crop in the context of strong climate change, with the possibility to develop the cultivation of sea fennel and increase its daily uses [2]. Often present as a wild edible halophyte plant, it can be used in the preparation of fresh salads, soups, and be minimally processed in a vinegar infusion [4]. Marinated fresh leaves and young branches with green olives have been the components of the diet for a long time (personal experiences). Sea fennel is known as a source of proteins, amino acids, vitamin C, minerals (potassium, sodium calcium, and magnesium), phenolic compounds, and flavonoids [5]. Dried SF has been used for aromatic traits, as well as food coloring like natural colorants [6]. It is used in traditional medicine as an anti-scorbutic, tonic, carminative, and diuretic [7]. SF as a wild vegetable plant with different phenolic compounds is indicated as a source of antioxidants, namely chlorogenic and phenolic acids [8].

Essential oils content of sea fennel (SFEOs) make a pleasant flavor, characterized by aromatic notes of fennel, celery and citrus peel [9]. Sea fennel as a halophytic plant species has developed mechanisms of tolerance to salinity, with reduced Cl^−^ absorption and its lower concentration inside the leaves, which leads to an increase in the concentration of soluble sugars and proline [10].

In addition to being collected in nature, sea fennel can also be cultivated on saline soils or in hydroponics [2]. Production in saline soils and conditions of drought stress emphasize the importance of sea fennel because other types of vegetables have a limited possibility of growth and development under such conditions [11].

Sea fennel EO from Dalmatia (Croatia) contains limonene (58.37%), sabinene (26.46%), terpinene-4-ol (5.59%), and γ-terpinene (2.81%) as the dominant compounds [12]. The EO analysis of different aerial plant parts of sea fennel from Central Dalmatia in the full flowering stage confirmed the presence of 14 compounds, dominated by sabinene (from 42.55 to 51.47%) and limonene (from 36.28 to 43.58%) [13]. The major compounds from Algerian SF found were γ-terpinene (50.5%), thymol methyl ether (33.6%), and o-cymene (12.6%). In comparison to the available literature, this chemical composition is peculiar and different from the EO extracted from Mediterranean and Atlantic ecotypes [14]. Essential oils from sea fennel, with their antibacterial properties and antioxidant activity, have potential as a cosmetic/cosmeceutical products [15].

Sea fennel EO showed a strong antimicrobial activity against many germs like *Candida albicans* and *Staphylococcus aureus*. Sea fennel may be considered a promising food plant for the future as it contains bioactive natural substances that may be used as nutraceuticals or agro-food supplements to increase their shelf life [15].

Sea fennel EOs from the Mediterranean Sea have been well documented, whereas there are no reports on the EO content and components of SF populations from Montenegro. This is the first research study on the essential oil content and bioactive compounds in different plant parts of SF in the area of Boka Kotorska (Montenegro).

The present paper aimed to compare the yield, chemical composition, and antioxidant activity of the essential oils isolated from different plant parts of SF grown in Montenegro.

## 2. Results and Discussion

### 2.1. Sea Fennel Essential Oil Content (SFEO)

SFEOs are determined by different factors besides their genotype and geographical distribution, including variable abiotic conditions, growth and stage of development, season differences, plant part, etc. The maximum SFEO was found in umbels with seeds (4.77 mL/100 g p.m.). The leaves contained less EO (0.52 mL/100 g p.m.) than immature inflorescence (0.83 mL/100 g p.m.) The minimum EO content was found in the stem (0.08%) (Table 1).

The content of the SFEOs varied depending on the plant part. Similar to our results, Generalić Mekinić et al. [16] revealed the content of EO in sea fennel and the differences in individual plant parts: the content of SFEO in the flowers was 2.44%, while the significantly lower amounts were isolated from leaves and stems, at 0.55 and 0.19%, respectively. The essential oils isolated by hydrodistillation from the aerial parts of sea fennel (*Crithmum maritimum* L.) were found to be a yellow liquid, with a yield of 0.2% (vol/wt) obtained based on wet weight [17]. The amount of essential oils in sea fennel reached about 0.8% in fruits, and 0.15 to 0.3% in the leaves [18]. The content of essential oil was found to be 0.6% [19].

Similar to our research, Houta et al. [20] reported that EO yields ranged between 0.28% and 3.60%. The highest amount was obtained from the sea fennel flowers (1.35%), while more than 2-fold lower amounts were detected in the stems and leaves [21]. The content of leaf SFEO from Turkey was found to be 0.6%, but the content of seed SFEO ranged between 8.39 and 11.66% [22].

The difference in the environmental factors and period of harvesting of the plants affect the EO content. The variation in the EO composition due to the different period of plant harvesting has been investigated previously [23,24].

### 2.2. Sea Fennel Essential Oil (SFEO) Composition

Twenty components were isolated from SFEO leaves, twenty-four from inflorescence, thirty-four components from the stem, and twenty-one components from umbels with seeds.

Limonene (62.4–72.0%), γ-terpinene (9.5–14.0%), α-pinene (1.4–5.8%), and sabinene (1–6.5%) were found to be the main components of the SFEO from monoterpene hydrocarbons as the dominant grouped components (86% to 98.1%), as shown in Table 2. Sea fennel plant parts showed differences in chemical profiles, especially in specific and low-represented ingredients. (*E*)-anethole (4.4%), fenchone (0.5%), and *trans*-carveol (0.2%) were present only in umbels with seeds, while the β-longipipene (0.5%), (*E*)-caryophyllene (0.5%), and (2*E*)-decenal (0.2%) were found only in the stems (Table 2).

The content of individual components of SFEO in our research differs depending on the plant part. Thus, compared to other plant parts, SF leaves contain the most limonene (72%) and sabinene (6.5%). Umbels with seeds are characterized by the highest content of alpha-pinene (5.8%) and (E)-anethole (4.4%) compared to other plant parts, while the stem contains the most dill apiole (4.7%) and carvacrol, methyl ether (3.3%), as indicated in Table 2.

The essential oil composition of SF grown on different parts of the Mediterranean coast showed differences in chemical constituents, suggesting the different chemotypes of this species [13]. Similarly to our research, SFEO from Croatia contained limonene as one of its main components. Limonene (58.37%), sabinene (26.46%), terpinene-4-ol (5.59%), and γ-terpinene (2.81%) were identified in wild sea fennel at the flowering stage as dominant compounds in essential oil from Croatia [12]. Another research group from Croatia presented similar results, where by sabinene (from 42.55 to 51.47%) and limonene (from 36.28 to 43.58%) were the main constituents of SFEO [13].

Sabinene comprised the highest percentage (43.29%) of sea fennel seed essential oil composition from Turkey (43.29%) [22]. Also, Baser et al. [22] confirmed that the main components of SFEO from Turkey were sabinene (26.9%), with a high content of limonene (24.2%) and γ-terpinene (19.3%). From another part of the Turkish coast, the EO mainly had monoterpenes, γ-terpinene (36% and 32%), β-phellandrene (21% and 22%), and sabinene (13% and 9%) as the main components [17]. According to previous research and literature citations, there are two types of SFs based on the main constituents in EO. The first ecotype is originally from the Atlantic area (France, Portugal, etc.) and the second from the Mediterranean (Croatia, Montenegro, etc.). The ecotypes from the Atlantic are abundant in dillapiol (62.10%) and carvacryl methyl ether, while the ecotypes from the Mediterranean are dominated by ingredients such as sabinene and limonene. Dillapiol is considered an undesirable ingredient in SF, and based on its content, the systematization and characterization of ecotypes was carried out. Dillapiol was absent from the SF samples from Croatia, giving it a special value [26]; on the other hand, it was present in different percentages in the samples from Turkey [27], Tunisia [28], Greece [29], France [30], and Portugal [31].

The chemical analysis of SFEO from Portugal revealed that γ-terpinene (33.6%), sabinene (32.0%), and thymol methyl ether (15.7%) are the major compounds [31]. The composition of essential oils depends on the climatic conditions during the vegetation season. SFEO samples from two different years in the same location show some differences inmain components [29].

It is interesting to point out that although Montenegro and Dalmatia (Croatia) are very close, neighboring countries, there are differences in the content of SF certain constituents. Although both varieties belong to the Mediterranean chemotype, in addition to the main constituents (sabinene and limonene) that are characteristic of SF in both countries, the content of dillapiol is expressed in the chemotype of SF from Montenegro, while it is never present in the chemotype of SF from Croatia [26].

The differences in synthesis, distribution, and accumulation of secondary metabolites in SFOEs is conditioned by abiotic (climatic, edaphic and orographic factors) and biotic factors (mutual relations of individuals, biological specificity, competitors),as well as time, intensity, and the method of harvesting, geographical origin and location, differences during the phenophases of development, differences in certain plant parts, and procedures after harvesting.

### 2.3. Antioxidant Activity (AA)–DPPH Assay

As an adaptation to overcoming strong environmental conditions such as high salinity, halophytes create and develop a defense mechanism based on their ability to produce secondary metabolites with antioxidant properties [32].

The degree of 2,2-diphenyl-1-picrylhydrazylradical (DPPH) neutralization increased with incubation time. The sea fennel EO isolated from stems showed stronger antioxidant activity during the incubation times of 20 and 40 min (EC_50_ value of 5.30 mg/mL and 5.04 mg/mL, respectively) in comparison to the SFEO isolated from other plant parts. The lowest antioxidant activity was obtained with the SFEO leaves (155.25 mg/mL and 58.30 mg/mL, respectively), as indicated in Figure 1 and Table 3.

The half-maximal inhibitory concentration (EC_50_) value reveals the antioxidant activity of essential oil: the lower the EC_50_ value, the higher the antioxidant activity. The EC_50_ value decreased with incubation time, and antioxidant activity, in fact, increased. The value of EC_50_ in the leaves and inflorescence decreased three times or twice from 20 to 40 min of incubation.

This study indicates that SFEO possesses significant antioxidant activities and is an important component in the food and pharmaceutical industries.

Sea fennel seeds (methanolic extract) are characterized by the highest DPPH· scavenging ability, with the lowest EC_50_ value (0.40 mg/mL). In the leaves, flowers, and stems, a slightly lower antioxidant activity (AA) was observed (EC_50_ values of 0.50, 0.70, and 0.72 mg/mL, respectively) [11].

A high content of phenolic compounds affects the stronger antioxidant activity of SF seed extract [8,32]. Due to the high AA, SF extracts can be widely used in food technology in the healthiness and functionality of food [33]. In addition to extracts, EOs are characterized by significant AA, but with a slightly lower activity. The DPPH activity of the EO was 2.8, 2.8, and 2.6% for flowers, stems, and leaves, respectively, while the DPPH (scavenging ability of the ethanolic extracts) was 61.0, 13.0, and 61.8% for flowers, stems, and leaves [16].

SFEOs with monoterpene hydrocarbons, limonene, and sabinene as major compounds did not have a radical scavenging effect on DPPH. It is known that the radical scavenging activity is determined by the number and substitution scheme of the -OH group and such compounds because they are not present in the tested oils, and a weak antioxidant activity of SF is to be expected.

Shahat et al. [24], in a study with three chemotypes of SF, stated that the antioxidant activity of EO (IC_50_ values determined by a DPPH assay) vary from 0.35 to 15.33 g/L and have relatively higher antioxidant activities than FSEO, in contrast to our results.

Different tests, such as DPPH (2,2-diphenyl-1-picrylhydrazyl), ABTS (2,2′-azino-bis-3-ethylbenzothiazoline-6-sulfonic acid), ORAC (oxygen radical absorbance capacity), and FRAP (ferric reducing antioxidant power), were used to evaluate the antioxidant properties of sea fennel extracts. Accordingly, Souid et al. [34] described the strong antioxidant activity from sea fennel leaves for DPPH (0.22 IC_50_ mg mL^−1^), ABTS (2.07 mg Trolox equivalents g^−1^), ORAC (15.84 μmol Trolox equivalents g^−1^ DW), and FRAP (1.82 EC_50_ mg mL^−1^).

Unlike the EO of sea fennel, the EO of common fennel in our previous research shows the highest antioxidant activity in the leaves. EOs from different parts of the wild fennel plant after incubation for 60 min show different AAs; leaves (12.37 mg/mL) > umbels first stage (20.52 mg/mL) > umbels second stage (29.89 mg/mL) > umbels third stage (31.97 mg/mL) > seeds (37.20 mg/mL). With the extension of the incubation time from 20 to 60 min, the degree of DPPH radical neutralization also increases [35]. Other factors like climatic conditions, region, time of harvesting, extraction method (temperature, time, etc.), and the polarity of the solvents used could also affect the antioxidant activity. Based on our results and literature references, SFEO exhibits antioxidant potential. Further research could lead to the inclusion of new methods of cultivation, as well as different applications of EO in the food industry and the preservation of fresh fruits and vegetables. Sea fennel could be grown in hydroponics and aqueous solutions instead of different substrates. The cultivation of this plant species is limited due to alack of cultivation technology; therefore, new knowledge and technical solutions are needed that would contribute to the faster spread of this very important vegetable species that could be an alternative to many vegetable and industrial crops, especially in soils that are salty, poor in nutrients, and short in irrigation water. Our analysis underscores the considerable phytochemical potential of sea fennel, suggesting opportunities for expanded cultivation and heightened consumer interest in this traditional type of vegetable species which is somewhat forgotten in Montenegro, while in Serbia, it would be a relatively novel vegetable species. The findings advocate for a greater awareness of the nutritional benefits associated with sea fennel, encouraging their incorporation into dietary practices and promoting sustainable cultivation practices for enhanced agricultural diversity.

## 3. Materials and Methods

### 3.1. Plant Material

Wild sea fennel was collected from the Herceg Novi seaside area (with coordinates of 42°27′05″ N 18°32′13″ E) during the vegetation period in2022 and 2023, identified by Prof. Dr. Zoran Ilić of the University of Priština in Kosovska Mitrovica. A voucher specimen has been deposited at the Herbarium of the Department of Biology in the Faculty of Agriculture. The samples when umbels reached physiological maturity and seeds started to break out (see the pictures in Appendix A).

The samples of sea fennel leaves, stems, inflorescences, and umbels with seedswere collected in September by hand or scissors. The different parts of sea fennel plants were dried in the shade (20 °C and 60–70% relative humidity). The dry material was ground to a fine powder before the extraction of essential oil.

### 3.2. Isolation of Essential Oil

Ground and homogenized sea fennel plant materials (stems, inflorescences, leaves, and umbels with seeds) were used for EO isolation by Clevenger-type hydrodistillation with a hydromodulus (ratio of plant material to water) of 1:10 m/V for120 min [36].

### 3.3. Gas Chromatography-Mass Spectrometry (GC/MS) and Gas Chromatography-Flame Ionization Detection (GC/FID) Analyses

The GC/MS and GC/FID analyses used are provided by Stanojević et al. [37]. GC/MS analysis was performed on an Agilent Technologies 7890B gas chromatograph, equipped with a nonpolar silica capillary column, HP-5MS (5% diphenyl- and 95% dimethyl-polysiloxane, 30 m × 0.25 mm, 0.25 μm film thickness; Agilent Technologies, Santa Clara, CA, USA), and coupled with an inert, selective 5977A mass detector of the same company. The essential oils obtained were dissolved in diethyl ether. One μL of the solution prepared was injected into the GC column through a split/split less inlet set at 220 °C in a 40:1 split mode. Helium was used as the carrier gas at a constant flow rate of 1 mL/min. The oven temperature increased from 60 °C to 246 °C at a rate of 3 °C/min. The temperatures of the MSD transfer line, ion source, and quadruple mass analyzer were set at 300 °C, 230 °C, and 150 °C, respectively. The ionization voltage was 70 eV and the mass range was *m*/*z* 41–415.

GC/FID analysis was carried out under identical experimental conditions as GC/MS. The flows of the carrier gas (He), make up gas (N_2_), fuel gas (H_2_), and oxidizing gas (Air) were 1, 25, 30, and 400 cm^3^/min, respectively. The temperature of the flame-ionization detector (FID) was set at 300 °C. Data processing was performed using MSD Chem Station F.01.00.1903 Mass Hunter Qualitative Analysis Version B.06.00 and AMDIS_32 software Version 2.70 (Agilent Technologies, USA). The retention indexes of the components from the analyzed samples were experimentally determined using a homologous series of n-alkanes from C_8_-C_20_ as the standards. Essential oil constituent identification was based on the comparison of their retention indexes (RI^exp^) with those available in the literature [25]. (RI^lit^), their mass spectra (MS) with those from Willey 6, NIST2011, and RTLPEST3 libraries, and, wherever possible, by co-injection with an authentic standard (Co-I). Semi-quantitative analysis, given as content expressed in percentage, was carried out using the area normalization method of the GC/FID signal without corrections.

### 3.4. Antioxidant Activity (AA)–DPPH Assay

The ability of the EO to scavenge free DPPH radicals was determined using the DPPH assay. The details of the method used are provided by Stanojevic et al. [36]. The DPPH test was used to determine the capacity of the pigment extracts (non-polar) and ethanolic extracts (polar) from sweet potato leaves to scavenge free 1,1-diphenyl-2-picrylhydrazyl (DPPH) radicals. The scavenging capacity was calculated from the equation of Stanojević et al. [37]. From the curve of dependence of the calculated DPPH scavenging capacities and concentrations of extract ethanolic solution, the concentration of the extract needed for neutralization of 50% of DPPH radical–EC_50_ (mg/mL) was determined. BHT was chosen as the positive control. Absorbance values (control, sample, and blank) were used for calculations regarding the degree of DPPH radical neutralization. All experiments were triplicates [38]. Although the most commonly used synthetic antioxidant BHT showed better antioxidant activity compared to both oils, it exhibits harmful effects in the human body [38].

### 3.5. Statistical Analysis

The difference between the means of sea fennel plant part essential oil yield was calculated with a T-test, while for other comparisons, an ANOVA was used. In the case of sea fennel yield, a one-way ANOVA was used, while for EC_50_, a factorial ANOVA was used. TIBCO Software Inc. (Palo Alto, CA, USA) (2020) Data Science Workbench, version 14. (http://tibco.com, accessed on 1 December 2020), was used to perform all statistical calculations.

## 4. Conclusions

Wild SF grows spontaneouslyon the Montenegrin cost represent an excellent source of biomolecules with important nutritional and antioxidant properties, characterized by certain specificities that make it different from the chemotypes in the surrounding and Mediterranean countries. The highest EO content was found in umbels with seeds (4.77 mL/100 g p.m.), while the stem contained the lowest EO content (0.08 mL/100 g p.m.). Limonene (62.4–72.0%) and γ-terpinene (9.5–14.0%) were the predominant constituents of SFEO from all plant parts. The total antioxidant activity was in the following order: stems (EC_50_ value of 5.30 mg/mL and 5.04 mg/mL) > inflorescences > umbels with seeds > leaves (155.25 mg/mL and 58.30 mg/mL). Therefore, the sea fennel chemotype from Montenegro is specific, and as such, it should be preserved and registered in the plant gene bank as a specificity, characterized, evaluated, and then presented with its characteristics and included in the diet in different forms, e.g., fresh, marinated, dried, as a bio product, with high antioxidant activity, as well in conditions of food shortage and survival, but also in traditional medicine and as a good antioxidant as a raw material for the cosmetic industry. At the same time, rich in bioactive compounds, SFEO may be an alternative to synthetic antioxidants in the food and pharmaceutical industries. The rational use of wild sea fennel, as well as the beginning of its cultivation on substrates or hydroponics would be the goal in the coming period. In the future, it will be necessary to investigate the seasonal variability in nutrient and essential oil content, and the palatability and antinutritive components of the different plant parts of SF.

## Figures and Tables

**Figure 1 plants-13-02003-f001:**
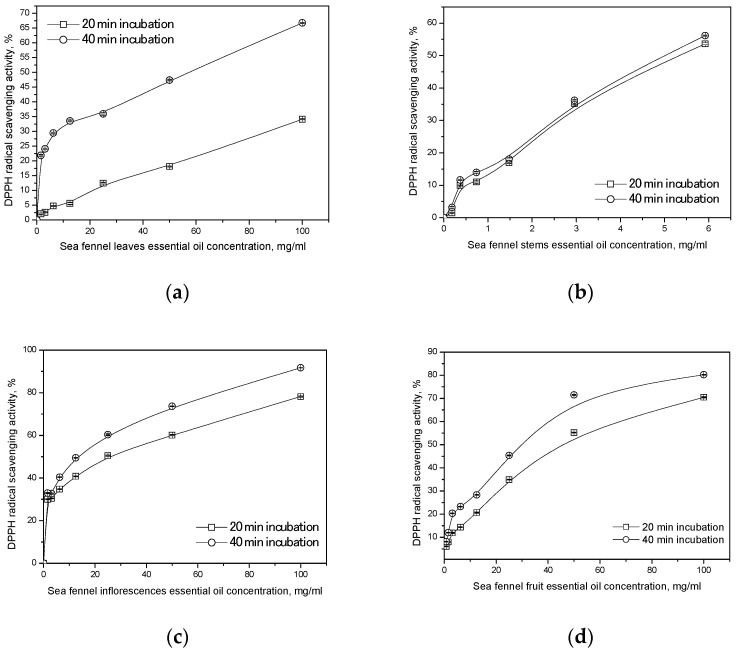
The degree of 2,2-diphenyl-1-picrylhydrazyl(DPPH) radical neutralization by sea fennel leaf (**a**), stem (**b**), inflorescence (**c**), and umbel with seed (**d**) essential oil.

**Table 1 plants-13-02003-t001:** Essential oil content from different sea fennel (SFEO) plant parts.

Plant Part	mL/100 g p.m.
Inflorescences	0.83 ± 0.008 ^b^
Leaves	0.52 ± 0.008 ^b^
Stems	0.08 ± 0.004 ^a^
Umbel with seeds	4.773 ± 0.106 ^c^

Values followed by different letters are significantly different at *p* < 0.05.

**Table 2 plants-13-02003-t002:** Chemical composition of sea fennel essential oil from the stem, inflorescence, leaves, and umbels with seeds.

N^o^	*t*_ret_, min	Compound	RI^exp^	RI^lit^	Method of Identification	Content %			
						Stem	Inflore Scence	Leaf	Umbels with Seeds
1.	6.44	α-Thujene	924	924	RI, MS	0.2	0.3	0.2	0.1
2.	6.65	α-Pinene	931	932	RI, MS, Co-I	1.4	3.9	2.7	5.8
3.	7.09	Camphene	946	946	RI, MS	tr	0.1	tr	0.1
4.	7.83	Sabinene	972	969	RI, MS	3.6	4.3	6.5	1.0
5.	7.94	β-Pinene	976	974	RI, MS, Co-I	tr	0.4	0.3	0.4
6.	8.34	Myrcene	990	988	RI, MS	1.0	1.1	1.1	0.6
7.	8.76	*n*-Octanal	1003	998	RI, MS	0.6	0.2	0.2	tr
8.	8.86	α-Phellandrene	1006	1002	RI, MS	-	-	-	tr
9.	9.26	α-Terpinene	1017	1014	RI, MS	0.3	0.3	0.2	tr
10.	9.60	*p*-Cymene	1026	1020	RI, MS	2.0	1.2	1.0	2.5
11.	9.79	Limonene	1031	1024	RI, MS, Co-I	62.4	68.6	72.0	66.2
12.	9.99	(*Z*)-β-Ocimene	1037	1032	RI, MS	3.9	3.2	4.4	2.0
13.	10.38	(*E*)-β-Ocimene	1047	1044	RI, MS	tr	tr	tr	-
14.	10.83	γ-Terpinene	1059	1054	RI, MS	11.3	14.0	9.5	13.5
15.	11.28	*cis*-Sabinene hydrate	1071	1065	RI, MS	tr	-	-	-
16.	11.93	Terpinolene	1088	1086	RI, MS	tr	tr	tr	-
17.	11.97	Fenchone	1089	1083	RI, MS	-	-	-	0.5
18.	12.06	*p*-Cymenene	1092	1089	RI, MS	tr	-		-
19.	12.54	*n*-Nonanal	1104	1100	RI, MS	0.5	tr	tr	-
20.	13.44	*cis*-*p*-Mentha-2,8-dien-l-ol	1126	1133	RI, MS	0.4	tr	tr	-
21.	13.96	*trans*-Limonene oxide	1138	1137	RI, MS	tr	-	-	tr
22.	14.87	(2*E*)-Nonen-1-al	1160	1157	RI, MS	tr	-	-	-
23.	15.76	Terpinen-4-ol	1181	1174	RI, MS	1.6	0.8	0.6	0.1
24.	16.41	α-Terpineol	1196	1186	RI, MS	tr	tr	-	tr
25.	16.80	*trans*-Carveol	1206	1215	RI, MS	tr	-	-	0.2
26.	17.64	*cis*-Carveol	1226	1226	RI, MS	tr	-	-	-
27.	17.83	Thymol, methyl ether	1230	1232	RI, MS	tr	-	-	-
28.	18.05	Carvacrol, methyl ether	1235	1241	RI, MS	3.3	1.4	0.7	-
29.	19.19	(2*E*)-Decenal	1262	1260	RI, MS	0.3	-	-	-
30.	20.12	Isobornyl acetate	1284	1283	RI, MS	-	tr	-	-
31.	20.43	(*E*)-Anethole	1291	1282	RI, MS, Co-I	-	-	-	4.4
32.	24.64	β-Longipipene	1392	1400	RI, MS	0.5	-	-	-
33.	25.73	(*E*)-Caryophyllene	1419	1417	RI, MS, Co-I	0.5	-	-	-
34.	28.28	ar-Curcumene	1482	1479	RI, MS	tr	-	-	-
35.	29.26	β-Bisabolene	1506	1505	RI, MS	0.6	tr	tr	-
36.	29.88	β-Sesquiphellandrene	1522	1521	RI, MS	0.3	tr	-	-
37.	31.15	Germacrene B	1556	1559	RI, MS	0.7	tr	-	0.2
38.	33.82	Dillapiol	1626	1620	RI, MS	4.7	0.3	0.5	2.5
					Total (%)	100.0	100.0	100.0	100.0
	Grouped components (%)								
	Monoterpene hydrocarbons (1–6, 8–13, 15, 16)					86.0	97.4	98.1	92.2
	Oxygen-containing monoterpenes (14,18,19,21–26)					5.3	2.1	1.3	0.6
	Sesquiterpene hydrocarbons (28–33)					2.6	tr	tr	0.2
	Phenylpropanoids (34)					4.7	0.3	0.5	7.0
	Others (7, 17, 20, 27)					1.4	0.2	0.2	tr

*t*_ret._: retention time; RI^lit^: retention indexes from the literature (Adams, 2007) [25]; RI^exp^: experimentally determined retention MS: constituent identified by mass-spectra comparison; RI: constituent identified by retention index matching; Co-I: constituent identity confirmed by GC co-injection of an authentic sample; tr = trace amount (<0.05%).

**Table 3 plants-13-02003-t003:** EC_50_ values of sea fennel essential oil from the different plant parts.

Plant Part of Sea Fennel (*Crithmum maritimum* L.)	EC_50_, mg/mL (20 min Incubation)	EC_50_, mg/mL (40 min Incubation)
Leaves	155.25 ± 1.287 b	58.30 ± 0.515 b
Stems	5.30 ± 0.020 a	5.04 ± 0.030 a
Inflorescences	28.22 ± 0.518 a	14.53 ± 0.144 a
Umbels with seeds (fruits)	50.49 ± 0.263 a	30.59 ± 0.060 a
Plant part	**	
Incubation	**	
Plant part × incubation	NS	

a,b: numbers in column marked with same letter are not significantly different (at 0.05 levels). **: Significance of ripening stage (significance at 0.01). NS: No Significance.

## Data Availability

The data presented in this study are available.

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
