# Peer review of "Essential Oil Content, Composition and Free Radical Scavenging Activity from Different Plant Parts of Wild Sea Fennel (Crithmum maritimum L.) in Montenegro"

_plants, 2024, doi:10.3390/plants13142003_

Round 1

Reviewer 1 Report (New Reviewer)

Comments and Suggestions for Authors

This study aims to compare yield, chemical composition and free radical scavenging activity of the essential oils from different plant parts of SF grown in Montenegro. This work is also interesting since the plant studied is resistant to climate changes. However, despite not being fluent in English, the text needs editing and improvements in the following points:

I suggest to altering the title “Essential Oil Content, Composition and Antioxidant Activity from Different Plant Part of Sea Fennel (Crithmum maritimum L.) Growing Wild in Montenegro” to “ Essential Oil Content, Composition and Free Radical Scavenging Activity from Different Plant Parts of Sea Fennel (Crithmum maritimum L.) Growing Wild in Montenegro”

The introduction section should be improved, the chemical composition of the SF essential oil (volatile compounds) and concomitant biological activities should not be mixed with the chemical composition of the plant. In general, essential oils have less antioxidant activity than their respective organic extracts.

Do not start sentences with abbreviations.

Please provide an image of the different parts of the sea fennel utilized in this study

Section 2.1. should be renamed to “Sea Fennel EO Content”

What do you mean by p.m.? is dry weight?

Dillapiol and dillapiole (or dill apiole) are the same compound? Please clarify.

Since DPPH is not a very reliable antioxidant method, could you perform ABTS instead?

Figure 1 should include the standard deviations.

Table 3 should display the statistical differences (ANOVA) and the standard results.

Line 201: The authors stated that “This study indicates that SFEO possess significant antioxidant activities…” however no standard assay was performed, and the best EC50 value was for the stem EO (5.03 mg/ml), which displayed the lowest yield and highest dillapiole content. Please discuss.

Expand abbreviations the first time they are mentioned, for example, "antioxidant activity (AA).

Include a paragraph in the discussion section about the study limitations.

Section 3.2. should be renamed to “EO isolation”. The hydrodistillation was performed 3 times?

Section 3.3. which standard (positive control) did you use?

Section 3.4 should be rewritten, as it does not match the title. Additionally, reference 40 is not correct.

Section 3.5. which software was used to perform the statistical analyses?

Author Response

I suggest to altering the title “Essential Oil Content, Composition and Antioxidant Activity from Different Plant Part of Sea Fennel (Crithmum maritimum L.) Growing Wild in Montenegro” to “ Essential Oil Content, Composition and Free Radical Scavenging Activity from Different Plant Parts of Sea Fennel (Crithmum maritimum L.) Growing Wild in Montenegro”

We accept your recommendation and create new title:  Essential Oil Content, Composition and Free Radical Scavenging Activity from Different Plant Parts of Sea Fennel (Crithmum maritimum L.) Growing Wild in Montenegro”

The introduction section should be improved, the chemical composition of the SF essential oil (volatile compounds) and concomitant biological activities should not be mixed with the chemical composition of the plant. In general, essential oils have less antioxidant activity than their respective organic extracts.

We try to reorganized Introduction part and improve them.  

Do not start sentences with abbreviations.

We change all sentence which start with abbreviations

Please provide an image of the different parts of the sea fennel utilized in this study

Sea fennel from Montenegro (org photo)

Section 2.1. should be renamed to “Sea Fennel EO Content”

We accept your suggestion

What do you mean by p.m.? is dry weight?

p.m. - plant material (dry weight)

Dillapiol and dillapiole (or dill apiole) are the same compound? Please clarify.

Dillapiol was correct

Since DPPH is not a very reliable antioxidant method, could you perform ABTS instead?

We haven’t enough quality of samples for ABTS methods

Figure 1 should include the standard deviations.

The values ​​of the standard deviation - error bars exist in the Figure 1, but they are very small and therefore difficult to see

Section 3.3. which standard (positive control) did you use?

BHT was chosen as positive control. Absorbance values (control, sample and blank) were used for calculations regarding the degree of DPPH radical neutralization. All experiments were triplicated (Milenković et al., 2023).

Although most commonly used synthetic antioxidant BHT showed better antioxidant activity compared

to both oils, it exhibits harmful effects in the human body (Milenković et al., 2023).

EC50, mg/ml

(20 min incubation)

EC50, mg/ml

(40 min incubation)

BHT

0.025 ± 0.001

0.023 ± 0.001

Milenković, A., Stanojević, J., Troter, D.,  Pejčić, M., Stojanović-Radić, Z.,  Cvetković, D., Stanojević Lj. (2023) Chemical composition, antimicrobial and antioxidant activities of essential oils isolated from black (Piper nigrum L.) and cubeb pepper (Piper cubeba L.) fruits from the Serbian market, Journal of Essential Oil Research, 35(3), 262-273.

Table 3 should display the statistical differences (ANOVA) and the standard results.

for EC50, factorial ANOVA was used. TIBCO Software Inc. (Palo Alto, CA, USA) (2020) Data Science Workbench, version 14. (http://tibco.com, accessed on 1 December 2020.), was used to perform all statistical calculations

Line 201: The authors stated that “This study indicates that SFEO possess significant antioxidant activities…” however no standard assay was performed, and the best EC50 value was for the stem EO (5.03 mg/ml), which displayed the lowest yield and highest dillapiole content. Please discuss.

Expand abbreviations the first time they are mentioned, for example, "antioxidant activity (AA).

We make correction

Include a paragraph in the discussion section about the study limitations.

We add some new references in discussion

Section 3.2. should be renamed to “EO isolation”. The hydrodistillation was performed 3 times?

We renamed to …..EO  isolation. (yes… 3 times)

Section 3.4 should be rewritten, as it does not match the title. Additionally, reference 40 is not correct.

We make correction

Section 3.5. which software was used to perform the statistical analyses?

The difference between means of sea fennel plant part essential oil yield was calculated with a T-test, while for other comparisons, ANOVA was used; in the case of sea fennel yield, one-way ANOVA was used, while for EC50, factorial ANOVA was used. TIBCO Software Inc. (Palo Alto, CA, USA) (2020) Data Science Workbench, version 14. (http://tibco.com, accessed on 1 December 2020.), was used to perform all statistical calculations.

We believe that our research is important because this is the first scientific publication in the area of ​​Montenegro and Serbia that investigates essential oils and their composition as well as antioxidant activity of sea fennel. It is also important that all plant parts of sea fennel that can be used have been processed individually. The significance lies in highlighting this plant species as a dietary option that our ancestors knew well, especially in the years of scarcity and survival. The richness of the biodiversity of the coast of Montenegro and the sea fennel that grows wild should be diversified and the wider public should be informed about its importance.

We try to improved scientific quality of MS include revision before being considered for publication

We would like to thank the Editor and Reviewer for their comments and suggestions to improve our manuscript. We hope that in current form our manuscript will meet high standards for publishing in Plants MDPI

Sincerely Yours

Guest Editors

Prof dr Zoran Ilic

Reviewer 2 Report (New Reviewer)

Comments and Suggestions for Authors

My comments on the manuscript are in the attached file.

Author Response

First, there are unclear and inconsistent statements in the text. For instance, in Line 41, "on our cost" should be clarified to "on Montenegro's cost."

Yes we adopt your suggestion "on Montenegro's cost."

Additionally, the abbreviations used in the paper should be employed consistently throughout, such as SFEO (Section 2.1) and SF (Line 105).

We add abbreviations

Furthermore, the title of Section 2.1 should be revised to enhance clarity.

We revised title…….. 2.1. Sea fennel essential oils content (SFEOs)

Moreover, there are formatting issues that need to be addressed. For example, the alignment of item 26 in Table 2 requires correction.

We make formatting in all Tables

 The paper must also be checked for editorial errors, such as the inconsistent citation numbering (e.g., Line 217, where citation 27 should be 24), to ensure readability.

We corrected references (27 with 24)

Furthermore, some references and discussions do not accurately align with the topic.

 Items 33, 34, and 35 in Lines 196-199 do not pertain to phenols in essential oils and should be omitted from the analysis.

We exclude these references

 Additionally, Line 146 references 2 does not refer to the analysis of EO by Croatian authors, only Italian authors. These inaccuracies must be corrected to ensure factual accuracy.

We corrected …….now stay ….

Limonene (58.37%), sabinene (26.46%), terpinene-4-ol (5.59%), and γ-terpinene (2.81%) were identified in wild sea fennel at the flowering stage as dominant compounds in essential oil from Croatia [12]. Another research group from Croatia presented similar results where sabinene (from 42.55 to 51.47%) and limonene (from 36.28 to 43.58%) as the main constituents of SFEO [13].

The manuscript would significantly improve with the addition of more recent references to bolster the findings and provide a modern context.

Different tests, such as DPPH (2,2-Diphenyl-1-picrylhydrazyl), ABTS (2,2′-azino-bis-3-ethylbenzothiazoline-6-sulfonic acid), ORAC (Oxygen Radical Absorbance Capacity), and FRAP (Ferric Reducing Antioxidant Power), are used to evaluate the antioxidant properties of sea fennel extracts. Accordingly, Souid et al. [] described the strong antioxidant activity from sea fennel leaves for DPPH (0.22 IC50 mg mL−1), ABTS (2.07 mg Trolox equivalents g−1), ORAC (15.84 μmol Trolox equivalents g−1 DW) and FRAP (1.82 EC50 mg mL−1).

Souid, A.; Della Croce, C.M.; Frassinetti, S.; Gabriele, M.; Pozzo, L.; Ciardi, M.; Abdelly, C.; Ben Hamed, K.; Magné, C.; Longo, V. Nutraceutical potential of leaf hydro-ethanolic extract of the edible halophyte Crithmum maritimum L. Molecules. 2021, 26, 5380.

Additionally, the reference sequence ought to be verified and corrected, as demonstrated in Line 207, where the sequence should be 8,33.

We corrected [8,33].

Section 3.1 is lacking information on the individual or entity responsible for designating the species and the location where the voucher specimen is stored. This essential data is vital for the scientific validity and traceability of the study.

Wild sea fennel was collected in Herceg Novi seaside area (with coordinates of 42°27′05″N 18°32′13″E) from the vegetation period during 2022 and 2023, identified by Prof. Dr. Zoran Ilić of the University of Priština in Kosovska Mitrovica. A voucher specimen has been deposited at the Herbarium of the Department of Biology in the Faculty of Agriculture.

Acronyms such as AA (Line 205) ought to be defined in the appropriate sections (e.g., Section 2.3) for improved comprehension.

We add explanation for acronyms AA (antioxidant activity).

A graphic abstract would have been a valuable addition to the reviewed manuscript.

We created graphic abstract

While the authors' conclusions are consistent with the results and presented effectively, addressing the aforementioned issues would improve the manuscript's clarity, accuracy, and overall quality.

We add part with the results……

The highest EO content was found in umbels with seed (4.77 mL/100 g p.m.).  while steam contained the lowest EO content (0.08 mL/100 g p.m.). Limonene (62.4-72.0 %) and γ-terpinene (9.5-14.0 %), are predominant constituents of SFEO from all plant part. Total antioxidant activity were in the order : stems (EC50 value of 5.30 mg/mL and 5.04 mg/mL) > inflorescent > umbel with seeds > leaves (155.25 mg/mL and 58.30 mg/mL.

We believe that our research is important because this is the first scientific publication in the area of ​​Montenegro and Serbia that investigates essential oils and their composition as well as antioxidant activity of sea fennel. It is also important that all plant parts of sea fennel that can be used have been processed individually. The significance lies in highlighting this plant species as a dietary option that our ancestors knew well, especially in the years of scarcity and survival. The richness of the biodiversity of the coast of Montenegro and the sea fennel that grows wild should be diversified and the wider public should be informed about its importance.

We try to improved scientific quality of MS include revision before being considered for publication

We would like to thank the Editor and Reviewer for their comments and suggestions to improve our manuscript. We hope that in current form our manuscript will meet high standards for publishing in Plants MDPI

Sincerely Yours

Guest Editors

Prof dr Zoran Ilic

Reviewer 3 Report (New Reviewer)

Comments and Suggestions for Authors

The present research evaluates the EA composition of different plant parts of fennels.

The introduction states the problem but should be reorganise, and the first reference should be changed.

The material and method has to be largely completed as many information are missing, both in the plant material section and in the analytical part, besides there is a mixture between the methods.

The results are well described globally, there are some minor corrections to be done. Those results are discussed with good references.

The conclusions should be reviewed as we are not looking for results in this section.

The abstract is correct.

Many comments and corrections could be found in the attached document

Main focus should be put on the Material and method section

Comments on the Quality of English Language

English verb tenses should be corrected to past in many situations as there is a mixture between present and past.

Some other minor corrections

Author Response

The present research evaluates the EA composition of different plant parts of fennels.

The introduction states the problem but should be reorganise, and the first reference should be changed.

Yes we are reorganized Introduction part

The material and method has to be largely completed as many information are missing, both in the plant material section and in the analytical part, besides there is a mixture between the methods.

We reorganized and added more details and information in Material and method part.

The results are well described globally, there are some minor corrections to be done. Those results are discussed with good references.

We make some minor correction

The conclusions should be reviewed as we are not looking for results in this section.

We revised Conclusions

The abstract is correct.

Many comments and corrections could be found in the attached document

We adopt all suggestion..and make correction ….follow remarks in Attached document

Main focus should be put on the Material and method section

We add and extend more information and detail in Material and method section

We believe that our research is important because this is the first scientific publication in the area of ​​Montenegro and Serbia that investigates essential oils and their composition as well as antioxidant activity of sea fennel. It is also important that all plant parts of sea fennel that can be used have been processed individually. The significance lies in highlighting this plant species as a dietary option that our ancestors knew well, especially in the years of scarcity and survival. The richness of the biodiversity of the coast of Montenegro and the sea fennel that grows wild should be diversified and the wider public should be informed about its importance.

We try to improved scientific quality of MS include revision before being considered for publication

We would like to thank the Editor and Reviewer for their comments and suggestions to improve our manuscript. We hope that in current form our manuscript will meet high standards for publishing in Plants MDPI

Sincerely Yours

Guest Editors

Prof dr Zoran Ilic

Round 2

Reviewer 1 Report (New Reviewer)

Comments and Suggestions for Authors

The manuscript has improved, it can be published

Author Response

Thanks for you suggestion !!!!!!!!

Reviewer 2 Report (New Reviewer)

Comments and Suggestions for Authors

 After completing the re-evaluation of the revised manuscript, in my opinion the authors have adequately addressed the concerns raised during the first round of peer review.

However, I would like to point out a few points that I think the authors should consider in the final proofreading of the manuscript before publication.

Once explained and used, the abbreviation for sea fennel essential oil (SFEO) or sea fennel (SF), for example, in the abstract, should be consistently used throughout the text.

Number 26 in Table 2 is still uncorrected.

The title of section 2.3. should include the abbreviation AA (for example 2.3. Antioxidant activity (AA) and this abbreviation should be used throughout the rest of the text.

The title of Section 3.4. should be consistent with the title of Section 2.3. The authors opted for a uniform title for both the sections.

Author Response

Thanks a lot for you support !!! We include your minor correction in last version of MS !!!!!!!

Reviewer 3 Report (New Reviewer)

Comments and Suggestions for Authors

The authors have provided an improved version of the initial document, where special attention has been paid to the introduction, which has been rearranged and now is much more fluent and coherent, but also material and method section has been completed with the data necessary to do again the reseach correctly.

The conclusion has been also completed.

The manuscrip is near to be ready to be submitted, the are minor errors to be corrected, particularly the number of decimals in two tables that must be uniform. Please find attached where you need to add a espace, and few other comments.

Author Response

thanks a lot for your support.

We accept  your recommendation  and include in last version.

Also we corrected tables 

This manuscript is a resubmission of an earlier submission. The following is a list of the peer review reports and author responses from that submission.

Round 1

Reviewer 1 Report

Comments and Suggestions for Authors

The authors addressed an important issue related to the use of substances of natural origin. The use of Essential Oils is gaining more and more attention, so research in the context of identification of EOs composition and potential methods of its use is of high significance. Authors presented the work related to Sea fennel Essential Oil (SFEO).

The scientific value of the work remains unclear to me. As authors indicated in lines 67-74, SFEO from Dalmatia and SFEO from Mediterranean and Atlantic were already reported and tested. I do not see any significant value in results showing only the composition of one SFEO from the area of Boka Kotorska in Montenegro. In my opinion, the scientific value of works consisting only of the composition of SFEO from subsequent regions and one simple assay for antioxidant activity is negligible. After identifying the ingredients included in particular SFEO, it can be concluded that they will have antioxidant activity. Moreover, it can be further concluded that the oil from each SFEO will have such an activity.

In my opinion, publication of results in a journal such as Plants requires, more results and a more thorough discussion of them. A comparison study of SFEOs of different region and more than just one assay on biological activity provides much more scientific value and allows authors to discuss obtained results e.g. in terms of comparison of content and biological activity of each EOs. From my perspective, combining the discussion section and the results section shows that it was difficult to find a topic to improve the discussion. Moreover, the work itself is too short for the “Article” type of work. Moreover, in my opinion the work is too short even for a communication (but I must admit that I did not count the words).

Please also note that the content of individual ingredients in SFEO may change due to the plant's growing conditions. Therefore, it is difficult to assess whether these results will be repeatable or not.

Author Response

 We believe that our research is important because this is the first scientific publication in the area of ​​Montenegro and Serbia that investigates essential oils and their composition as well as antioxidant activity of sea fennel. It is also important that all plant parts of sea fennel that can be used have been processed individually. The significance lies in highlighting this plant species as a dietary option that our ancestors knew well, especially in the years of scarcity and survival. The richness of the biodiversity of the coast of Montenegro and the sea fennel that grows wild should be diversified and the wider public should be informed about its importance.

Reviewer 2 Report

Comments and Suggestions for Authors

[Plants] Manuscript ID: plants-2972072

There are several typos, inconsistencies in the presentation of the text. For example:

1- Line 27: respectively.

2- Line 72: ǫ-cymene or o-cymene?

3- There is a need for space between digits and units. See lines 27-29, 190-192, 197-199 and 254.

4- There is a need for space at the end of text and before the reference. See line 41.

5- Generally, the references are found in the text in numerical order. This is not the case for references 16 to 18, which are found on lines 269-274 after reference 42 on line 223.

6- In the same vein, a reference 41 appears in line 113, immediately along with reference 26.

7- Line 134: (E)-anethole.

8- Line 254: 10-15°C or 10-15 °C?

9- Numbers appear as exponents. Or are they more about references? See lines 109, 211 and 265.

10- Line 119: Table 3 or Table 2?

11- Lines 243-244: The sentence: The plant materials were identified … would be better placed in the Authors' contribution section.

Table 3, line 32: -longipipene or -longipinene?

The list of references should be carefully reviewed:

1- In many cases, the reference does not include the issue number. This information would be very helpful to the reader.

2- Does the Journal want to see the web address after each reference?

3- Ref 2 and ref 10 are the same; idem for ref 3 and 31; idem for ref. 8 and 38; 32 and 39.

4- Line 312: and taste. Foods …

5- Line 316: Many uppercase letters should be lowercase letters.

6- Line 327: Machado, R.M.A.; Serralheiro, R.P.

7- Line 334: Is there a reason for the authors' underlining?

8- Line 341: Illić, Z.S. Lalević, D;.

9- Line 347: Burčul.

10- Line 358: Bratincević or Bratinčević?

11- Line 364: Crithmum maritimum.

12- Line 365: Many uppercase letters should be lowercase letters.

13- Line 370: there are several inappropriate ´c.

14- Line 399: Annona cherimola.

15- Line 406: 2023, 9, 364.

Author Response

Reviewer 2.

Please also note that the content of individual ingredients in SFEO may change due to the plant's growing conditions. Therefore, it is difficult to assess whether these results will be repeatable or not.

There are several typos, inconsistencies in the presentation of the text. For example:

  • Line 27: respectively.

Yes we accept your suggestion

  • Line 72: ǫ-cymene or o-cymene?

o-cymene was correct

  • There is a need for space between digits and units. See lines 27-29, 190-192, 197-199 and 254.

We improve and make space between digits and units

  • There is a need for space at the end of text and before the reference. See line 41.

Yes we add space…..

  • Generally, the references are found in the text in numerical order. This is not the case for references 16 to 18, which are found on lines 269-274 after reference 42 on line 223.

              We did …now all reference stay in numerical order

  • In the same vein, a reference 41 appears in line 113, immediately along with reference 26.

References 41 now is 24 ..and stay immeaditely after ref 23

7- Line 134: (E)-anethole.

(E)-anethole was correct

8- Line 254: 10-15°C or 10-15 °C?

 Now stay 10-15 °C

9- Numbers appear as exponents. Or are they more about references? See lines 109, 211 and 265.

We remove all exponent references

10- Line 119: Table 3 or Table 2?

Table 2 was correctly

11- Lines 243-244: The sentence: The plant materials were identified … would be better placed in the Authors' contribution section.

We accept your suggestion and replace this in Authors contribution section.

-longipinene? b-longipipene or bTable 3, line 32:

β-Longipipene

The list of references should be carefully reviewed:

1- In many cases, the reference does not include the issue number. This information would be very helpful to the reader.

2- Does the Journal want to see the web address after each reference?

3- Ref 2 and ref 10 are the same; idem for ref 3 and 31; idem for ref. 8 and 38; 32 and 39.

We did……exclude duplicate references

4- Line 312: and taste. Foods …add space

5- Line 316: Many uppercase letters should be lowercase letters. Yes we make changes

6- Line 327: Machado, R.M.A.; Serralheiro, R.P. …..changed in  Machado, R.M.A.; Serralheiro, R.P

7- Line 334: Is there a reason for the authors' underlining? Its mistake

8- Line 341: Illić, Z.S. Lalević, D;.exclude this references

9- Line 347: Burčul… accept Burčul

10- Line 358: Bratincević or Bratinčević? Bratinčević

11- Line 364: Crithmum maritimum.   Add space

12- Line 365: Many uppercase letters should be lowercase letters. Changes all

13- Line 370: there are several inappropriate ´c……… we are improve

14- Line 399: Annona cherimola.   Latin name …Italic

15- Line 406: 2023, 9, 364.    Italic  9

Reviewer 3 Report

Comments and Suggestions for Authors

The topic of the work is quite interesting, especially that the plant is not cultivated on a wider scale. The experiment was carried out correctly, the results are promising and require confirmation in further research. Minor comments noted in the manuscript.

Comments on the Quality of English Language

The work is written correctly and legible

Author Response

We would like to thank the Editor and Reviewer for their comments and suggestions to improve our manuscript. We hope that in current form our manuscript will meet high standards for publishing in your journal Plants  MDPI

Sincerely Yours

Prof dr Zoran Ilic

Reviewer 4 Report

Comments and Suggestions for Authors

The aim of the study was to determine the content, composition and antioxidant activity of various parts of sea fennel. The presented research allowed to achieve the assumed goal.

The work requires only minor editorial corrections. Fragments to be corrected are marked in yellow in the original text (pdf) along with comments

Author Response

(The authors gave the same response as above.)

Round 2

Reviewer 1 Report

Comments and Suggestions for Authors

In their response to the review, the authors did not provide an explanation related to the scientific value of this work. I maintain the opinion from the previous review that the work is too short and its scientific value and significance are at an insufficient level. I would like to point out that the journal Plants has an impact factor of 4.5, and the reviewed work definitely does not meet the requirements for works published in this journal

Moreover, the article is rather regional rather than global, which leads to the assumption that it will not be cited by the authors of other works.

Author Response

We ask the reviewer for his understanding because there are young scientists who need afirmation throught publication of articles. It is up to us to give promise that we will continue to work on this plant species and enrich the research with the results of questions about the content of phenols, flavonoids Microbial properties of EO....and a number of important questions related to this wild halophyte vegetable plant  that can survive in very limited condition for growing. This is first research in Montenegro and Serbia on this species where all plants part are included and where the emphasis is on essential oils that can have multiple applications in food, pharmaceutical and cosmetics industries. The manuscript is short, but contains more of 4000 words which is above the lower limit for this type of article. I note that the do science in very simple condition in North Kosovo, publication of article gives us hope and faith in a better fure and survival even in small University. 

Once again thanks for your support and understanding.

Sincerely Yours 

Correstonding authors and Guest Editor ,

Prof dr  Zoran Ilic